# A Deep Learning Model Integrating Clinical and MRI Features Improves Risk Stratification and Reduces Unnecessary Biopsies in Men with Suspected Prostate Cancer

**DOI:** 10.3390/cancers17132257

**Published:** 2025-07-07

**Authors:** Emiliano Bacchetti, Axel De Nardin, Gianluca Giannarini, Lorenzo Cereser, Chiara Zuiani, Alessandro Crestani, Rossano Girometti, Gian Luca Foresti

**Affiliations:** 1Institute of Radiology, Department of Medicine (DMED), University of Udine, and University Hospital “Santa Maria della Misericordia”, ASU FC, P.le S. M. Della Misericordia 15, 33100 Udine, Italy; bacchetti.emiliano@spes.uniud.it (E.B.); lorenzo.cereser@uniud.it (L.C.); chiara.zuiani@uniud.it (C.Z.); rossano.girometti@uniud.it (R.G.); 2Artificial Vision and Machine Learning Laboratory (AVML Lab), Department of Mathematics, Computer Science and Physics (DMIF), University of Udine, Via delle Scienze 206, 33100 Udine, Italy; axel.denardin@uniud.it; 3Urology Unit, University Hospital “Santa Maria della Misericordia”, ASU FC, P.le S. M. Della Misericordia 15, 33100 Udine, Italy; 4Urology Unit, Department of Medicine (DMED), University of Udine, and University Hospital “Santa Maria della Misericordia”, ASU FC, P.le S. M. Della Misericordia 15, 33100 Udine, Italy; alessandro.crestani@uniud.it; 5Artificial Vision and Real-Time Systems Laboratory (AViReS Lab), Department of Mathematics, Computer Science and Physics (DMIF), University of Udine, Via Delle Scienze 206, 33100 Udine, Italy; gianluca.foresti@uniud.it

**Keywords:** prostate neoplasms, magnetic resonance imaging, artificial intelligence, biopsy

## Abstract

We developed three deep learning (DL) models for upfront risk stratification of clinically significant prostate cancer (csPCa) and reduction in unnecessary biopsies, using data from 538 Caucasian men who underwent Magnetic Resonance Imaging (MRI) and biopsy. Model 1 (M1) used clinical data only, Model 2 (M2) relied on the Prostate Imaging Reporting and Data System (PI-RADS) scores, and Model 3 (M3) combined both. Model 3 achieved the highest accuracy (Area under the curve [AUC] 0.822), reducing unnecessary biopsies by 43.4% compared to M1 and 21.2% compared to M2. Although M3 had slightly lower sensitivity than M2, it offered better specificity and positive predictive value. Decision curve analysis showed that M3 provided greater net benefit in biopsy-averse scenarios (higher disease probability), while M2 was more effective in cancer-averse settings (lower disease probability). These findings highlight the potential of DL models to support personalized, context-driven decisions in the early detection of csPCa.

## 1. Introduction

Over the past years, prostate magnetic resonance imaging (MRI) has become the reference imaging test for triaging patients for prostate biopsy [1]. With a 91% negative predictive value (NPV) [2], prostate MRI can prevent unnecessary biopsies in approximately 30% of men undergoing the examination, while posing a limited risk of missing high-grade cancers [3]. MRI can also guide prostate biopsy by targeting suspicious areas, thereby enhancing the detection of clinically significant prostate cancer (csPCa) compared to systematic biopsy alone [4]. On the other hand, prostate MRI shows a positive predictive value at patient-level as low as 40% in a meta-analysis [5], leading to a significant number of false-positives which, in turn, translate into unnecessary prostate biopsies. Additional factors affecting MRI as a stand-alone risk stratification tool include limited inter-reader agreement [6] and suboptimal sensitivity for local staging [7].

By combining readily available clinical and laboratory variables such as patient age, prostate-specific antigen (PSA) level, family history, and digital rectal examination (DRE), calculators have been extensively explored as tools for upfront risk stratification [8]. While calculators are effective in reducing unnecessary biopsies [9], they tend to perform better when MRI results are integrated in the model. For example, an external validation study of different calculators on 468 biopsy-naive men [10] showed that those incorporating MRI performed better in discriminating csPCa and guiding appropriate biopsy decisions compared to those without MRI. Thus, integrating MRI results into risk calculators represents a promising strategy for risk stratification and patient selection for biopsy, compared to using MRI alone [1,8].

Most available calculators have been built upon linear and logistic regression. The use of artificial neural networks has been advocated as a more robust approach in many medical imaging scenarios [11,12], including building prostate cancer risk models, compared to regression [13], in line with the potential of artificial intelligence-based systems to handle complex and diverse clinical data, offering improved patient stratification and personalized treatment strategies. To our knowledge, only a few models have utilized deep learning (DL) combined with PSA [14] or additional biomarkers [15] to stratify pre-biopsy patients’ risk, demonstrating increased specificity and reduced unnecessary biopsies. Little is known about risk calculators that incorporate more clinical variables and prostate MRI results. Parekh et al. [13] demonstrated that a model designed using a three-layered backpropagation ANN and incorporating age, ethnicity, digital rectal examination, family history, PSA, biopsy status, prostate volume, and MRI (Mount Sinai Prebiopsy Risk Calculator) yielded an 8–10% higher probability of accurately predicting clinically significant prostate cancer (csPCa) compared to conventional models. However, further research should support the use of DL-based models in this clinical setting, clarify whether DL-based models benefit from the inclusion of MRI results as conventional ones do, and determine which architecture performs best.

On this basis, the purpose of our study was to train and internally validate a DL-based model for predicting the risk of csPCa based on clinical, laboratory and MRI features. Additionally, we aimed at comparing its performance against a DL model not including MRI and against MRI alone. This article is organized into sections detailing the methodology, dataset construction, and model development, followed by validation results and discussion of clinical implications. The innovation of our study lies in applying deep learning for upfront risk stratification—rather than relying on traditional logistic regression models commonly used in non–DL-based calculators—and in leveraging structured MRI-derived data without requiring direct image analysis or the use of DL systems for interpreting MRI images. This approach enhances accessibility and usability, especially for clinicians who do not read MRI images or do not have access to AI tools for image interpretation.

## 2. Materials and Methods

### 2.1. Study Population and Standard of Reference

This study was approved by our Institutional Review Board. The requirement for informed consent from patients was waived due to the retrospective design. We included all men who, between April 2019 and September 2024, were consecutively referred for prostate MRI due to clinical suspicion of prostate cancer, indicated by a PSA level of ≥3 ng/mL in two serial samples and/or a positive digital rectal examination (DRE) and/or family history of prostate cancer. These men subsequently underwent prostate biopsy due to suspicious MRI findings or higher clinical risk despite a negative examination. After applying the exclusion criteria shown in Figure 1, the final population consisted of 538 men, whose characteristics are described below. Notably, all of them were of Caucasian ethnicity.

The standard of reference for csPCa was represented by prostate biopsy performed by local urologists under ultrasound guidance and software-assisted fusion with MRI images (Aplio 300 ultrasound platform, Canon Medical Systems, Ōtawara, Tochigi, Japan). The procedure was performed via the transperineal route, combining targeted biopsies of suspicious MRI findings (Prostate Imaging Reporting and Data System version 2.1 [PI-RADS] category ≥ 3 [16]) with a 12-core systematic biopsy. Target biopsies included two in-field and two perilesional cores. Histological analysis of the biopsy samples was performed by referring uropathologists. csPCa was defined as a Grade Group 2 lesion according to the International Society of Urological Pathology (ISUP) [17].

### 2.2. MRI Protocol and MRI Analysis

Prostate MRI was performed using a 1.5 T magnet (Aera, Siemens Healtheneers, Erlangen, Germany) or 3.0T (Achieva, Philips Healthcare, Eindhoven, The Netherlands) with a 32-element surface coil. Men underwent preliminary preparation, including a rectal enema and, if not contraindicated, intramuscular or intravenous administration of 20 mg butylscopolamine as an antiperistaltic agent. As detailed in Appendix A, the multiparametric protocol included high-resolution T2-weighted imaging acquired in the axial plane with at least one complementary sagittal and/or coronal plane, two separate diffusion-weighted sequences, and high-temporal resolution dynamic contrast-enhanced imaging (DCE).

MRI examinations were interpreted by a pool of radiologists using PI-RADS version 2.1. As readings were performed during clinical routine, radiologists were unblinded to clinical information. For the purpose of having a homogeneous dataset, a single radiologist meeting the requirements of an “expert reader” [18] retrospectively reviewed 40/538 MRIs originally reported by other non-expert readers, refining the PI-RADS category assigned to any lesions or identifying a new index lesion. The latter was defined as the lesion with the largest PI-RADS category or largest size and/or extraprostatic extension in the case of similar PI-RADS categorization. The absence of lesions in the prostate corresponded to PI-RADS 1 by definition. The originally attributed PI-RADS categories were those prompting (or not) prostate biopsy. In cases of discrepancy between the index lesion originally reported and the one identified and/or re-categorized by the expert reader, the diagnosis of csPCa was based on target biopsy results, if available on that target, or the nearest core of systematic biopsy.

### 2.3. Dataset Building and Definition of the Models

For each patient, the dataset finally included the following clinical, laboratory and imaging variables for building the model: (1) age; (2) PSA level at the time of prostate MRI (within 3 months before MRI); (3) PSA density calculated by dividing the PSA level by the prostate volume measured on MRI images according to PI-RADS v2.1 rules; (4) Previous history of negative prostate biopsies, if any; (5) Results of DRE (suspicious or non-suspicious); (6) Family history of prostate cancer, if any, as defined by EAU guidelines [1]; (7) Ongoing therapy with alpha-blockers or 5-alpha-reductase inhibitors, if any; (8) Prostate volume on prostate MRI; (9–11) Site (base, midgland, or apex), size and PI-RADS category of the index lesion as determined by the PI-RADS v2.1 rules [16].

Three distinct DL models were developed to calculate the confidence in diagnosing csPCa on a 0-to-1 scale, with model 1 (M1) relying exclusively on clinical variables (variables 1–7, as listed above), model 2 (M2) incorporating prostate MRI results as categorized by PI-RADS 2.1, and model 3 (M3) combining both clinical and MRI-related variables (variables 1–11, as listed above). M1 and M3 were developed using a deep learning approach to generate patient-level csPCa predictions.

### 2.4. Pre-Processing

Figure 2 illustrates the pipeline for the construction of the DL model. Before feeding the data to the model, a set of pre-processing steps was performed to ensure that the input data was in an optimal format for training. The first key step involved transforming the “lesion location” parameter, which was initially represented as a categorical variable with three possible values (base, mid, apex). Since machine learning models typically perform better with numerical representations, this categorical variable was split into three distinct binary parameters, each indicating the presence (1) or absence (0) of a lesion in the corresponding anatomical location. This transformation allowed the model to interpret lesion location information in a structured numerical format.

A crucial consideration in this transformation was the fact that these three binary variables were not mutually exclusive, meaning that a lesion could be present in multiple regions simultaneously. This flexibility ensured that the model accurately captured complex lesion distributions without imposing artificial constraints on the data.

In addition to restructuring categorical variables, all continuous numerical variables underwent a normalization process. Normalization was necessary to bring all input variables into a uniform scale, preventing parameters with inherently larger numerical values from disproportionately influencing the model’s learning process. This step was particularly important given the heterogeneous nature of medical imaging and clinical data, where different features can have vastly different ranges and distributions. Normalization also contributed to improved model performance by reducing training time and facilitating faster convergence during optimization.

### 2.5. Building, Training, and Testing

The model architecture was designed as a fully connected artificial neural network (ANN) with three hidden layers consisting of 64, 32, and 16 nodes, respectively. Each layer utilized the ReLU (Rectified Linear Unit) activation function, which is widely used for deep learning applications due to its ability to mitigate vanishing gradient issues and promote efficient learning.

To enhance the model’s stability and improve training efficiency, a batch normalization layer was introduced between network layers. Batch normalization helps regulate the distribution of activations, reducing internal covariate shift and enabling faster and more stable training. Furthermore, to mitigate overfitting and improve the generalization capability of the model, a dropout layer was incorporated with a dropout rate of 70%. This high dropout rate forced the network to learn more robust features by randomly deactivating a significant proportion of neurons during training, preventing over-reliance on specific features in the training set.

The model was trained using the Binary Cross-Entropy (BCE) loss function [19], which is particularly suitable for binary classification tasks such as distinguishing between cancerous and non-cancerous lesions.

It quantifies the discrepancy between the predicted probabilities output by the model and the ground-truth binary labels. For a single prediction, the BCE is defined as follows:LBCE=−[y⋅log(p)+(1−y)⋅log(1−p)]
where
y∈{0,1} is the ground truth label*p* ∈ (0,1) is the predicted probability of the positive class.

This formulation penalizes incorrect predictions more heavily as their confidence increases.

Optimization was performed using the Adam optimizer [20], known for its adaptive learning rate adjustments and computational efficiency. Adaptive Moment Estimation (Adam) is a widely used stochastic optimization algorithm that combines the advantages of two other methods: AdaGrad (adaptive learning rates) and RMSProp (momentum on the squared gradients). It is designed to efficiently handle sparse gradients and non-stationary objectives, which are common in deep learning. Adam maintains per-parameter learning rates that are adapted based on estimates of first and second moments of the gradients:The first moment is the exponentially decaying average of past gradients (akin to momentum).The second moment is the exponentially decaying average of the squared gradients.

Adam is favoured for its robust convergence, minimal hyperparameter tuning, and effective scaling across deep architectures, making it a default choice in many neural network training scenarios.

The initial learning rate was set to 0.01, and a weight decay of 1 × 10^−4^ was applied to prevent overfitting by introducing an additional regularization term.

Given the relatively small dataset available for training, a 5-fold cross-validation approach was implemented to ensure a robust evaluation of the proposed architecture. This method divided the dataset into five equal parts, where in each iteration, four subsets were used for training, and one was reserved for testing. The process was repeated five times, with each subset serving as the test set once. This approach minimized potential biases arising from a specific train-test split and provided a more reliable assessment of the model’s performance.

### 2.6. Data Analysis

Descriptive statistics were employed to characterize the study population and the variables incorporated into the models. Given that the Shapiro–Wilk test indicated a non-normal distribution of quantitative variables, median values and interquartile ranges (IQRs) were used for summarization. Relevant rates were reported with accompanying 95% confidence intervals (95% CI). Statistical comparisons between the training and test sets were not performed, as the prevalence of variables was observed to be relatively similar across both cohorts (see results below).

Each model was subjected to a calibration analysis, wherein predicted probabilities of csPCa were plotted against observed probabilities. Predicted probabilities were derived through univariable and multivariable analyses, employing a stepwise approach. Model discrimination was assessed via receiver operating characteristic (ROC) analysis by calculating the area under the curve (AUC) for each model and selecting confidence thresholds better approximating benchmark MRI sensitivity and specificity values of ≥90% and ≥50%, respectively. For M3, the selected sensitivity exceeded the lowest 95% confidence interval value of M2 sensitivity, while its specificity surpassed the highest 95% confidence interval of M2s specificity, establishing M2 as the benchmark for improvement. This refinement aimed to reduce false positives while maintaining a comparable ability to identify true positives. Positive Predictive Value (PPV) and Negative Predictive Value (NPV) were calculated by classifying each case as “positive” or “negative” according to whether the model output was above or below the confidence threshold identified through ROC analysis, respectively. The calculations followed the standard formulas as reported elsewhere [21]. AUCs for the three models were compared using the DeLong method [22]. Alfa level was set 0.05.

The impact of the models on biopsy decision-making was examined through decision curve analysis (DCA) [23]. Two reference strategies were considered: a treat-all approach (biopsying all men) and a treat-none approach (biopsying no men). The strategy yielding the highest net benefit was identified as the optimal balance between correctly diagnosing true-positive csPCa cases and minimizing unnecessary biopsies performed on false-positive cases.

Analysis was performed using MedCalc Statistical Software version 23.1.3 (MedCalc Software Ltd., Ostend, Belgium) and Stata version 19.5 (Stata Corp LLP, College Station, TX, USA).

## 3. Results

### 3.1. Study Population and MRI Results

Table 1 summarizes the clinical characteristics of the 538 men included in the study, along with MRI findings, prostate cancer histology, and staging based on imaging. The prevalence of csPCa was 35.3% (190/538 men).

### 3.2. Calibration and Discrimination

Appendix A presents the calibration plots for M1, M2, and M3. M1 exhibited a tendency to underpredict csPCa, whereas Model 3 (M3) demonstrated a closer alignment between predicted and observed probabilities along the diagonal reference line, albeit with a slight overall tendency toward overprediction. Model 2 (M2) tended to underpredict csPCa for PI-RADS categories 1–2 and 5, while overpredicting for PI-RADS categories 3–4.

Table 2 shows the results of logistic regression, while Table 3 presents a summary of the diagnostic performance metrics for assessing csPCa, along with the thresholds derived from ROC analysis and details on the grading group od missed cancers. Figure 3 displays the AUCs for the three models, showing that both M2 (*p* = 0.0188) and M3 (*p* < 0.0001) achieved significantly higher AUCs than M1. Additionally, the AUC for M3 was significantly higher than that of M2 (*p* = 0.0003). Miss rate for csPca was 12.6% for M1 (24/190), 8.4% for M2 (16/190) and 13.2% for M3 (25/190). Figure 4 shows the confusion matrices for M1, M2 and M3, while Figure 5 shows an MRI example case.

While M3 exhibited a slight decrease in sensitivity compared to M2, it improved specificity and PPV, reducing false-positive and, in turn, unnecessary biopsies by 100/230 (43.4%) and 35/165 (21.2%) compared to M1 and M2, respectively.

### 3.3. Clinical Impact

Figure 6 illustrates the DCA plots for the three models, while Table 4 presents the net benefit values at relevant threshold probability cut-offs. M2 demonstrated the highest net benefit for threshold probabilities ≤ 20%, while M3 exhibited the highest net benefit from >20% onwards.

## 4. Discussion

Our study showed that, at patient level, a DL-based model (M3) integrating clinical and MRI-derived variables significantly improved upfront risk stratification for csPCa by reducing false positives and, consequently, avoiding approximately 43% and 20% of unnecessary biopsies compared to a clinical-only model (M1) and an MRI-only model (M2), respectively. This reduction was achieved while maintaining sensitivity and NPV as close as possible to those of MRI alone (86.8% and 89.7% for M3 vs. 91.6% and 91.9% for M2), which represent a benchmark in the current scenario of early detection of csPCa in a population with 30–50% of disease prevalence (35.3% in our series) [24]. Notably, most of the missed cases were ISUP grade 2 cancers at the lowest edge of the spectrum of aggressiveness. While the relatively low AUCs of the models, including M3, still reflect room for further improvement in specificity and the unavoidable trade-off with sensitivity, we believe our work provides a research direction in which AI-based calculators can meaningfully enhance biopsy decision-making compared to the current clinical scenario.

DCA underscored the complementary utility of our models across varying clinical priorities. M2 achieved greater net benefit at threshold probabilities ≤ 20%, reflecting a cancer-averse scenario wherein the primary objective is minimizing missed csPCa diagnoses, even at the expense of increased biopsy rates. Conversely, M3 outperformed above this threshold, aligning with a biopsy-averse perspective that favors reducing unnecessary procedures while maintaining acceptable diagnostic sensitivity. These findings suggest that M2 and M3 may be selectively applied to support personalized decision-making, depending on the clinical scenario and the shared preference between patient and clinician [25]. Compared to a strategy of biopsying all patients, our M3 model achieved a substantially higher reduction in unnecessary biopsies (43%) than the clinical model only (M1). This is higher than previously reported for widely used MRI-based risk calculators such as the Van Leeuwen model and the RPCRC, which in external validation studies reported reductions around 20–33% at a 10% risk threshold compared to a “biopsy all” strategy [26,27]. While M3 also reduced unnecessary biopsies by approximately 20% compared to MRI alone (M2), this gain in specificity came at the cost of a slightly higher miss rate (13.2% vs. 8.4%). A potential explanation relies on higher decision threshold used for M3 (0.32 vs. 0.21 probability), which inherently leads to a greater specificity but lower sensitivity at that specific operational point, and may further indicate that the combined clinical-MRI model should be preferred in a biopsy-averse scenario, while a stand-alone MRI model suffices for cancer-averse scenarios.

Our findings align with a substantial body of evidence supporting the added value of integrating MRI-derived variables into clinical risk models [8,27,28]. However, the use of a DL-based approach makes our results more directly comparable with the previously mentioned Mount Sinai Prebiopsy Risk Calculator [13], which was developed using a large, prospectively collected cohort of 1902 men (2363 biopsies) and a set of variables including age, PSA, DRE, PI-RADS, biopsy history, ethnicity, family history, and prostate volume. Their model achieved an AUC of 0.92 for csPCa when tested with artificial neural networks, which exceeds the AUC of 0.882 observed for our M3 model. Differences in population characteristics and model architecture may account for the lower AUC, though comparable to the 0.82 reported by the authors when validating the MSP-RC in the PROMOD cohort. Differently from the Mount Sinai model, our study was inherently developed and tested for clinical utility within a DL architecture. Furthermore, we incorporated additional clinical variables such as PSAD and the use of medications known to affect PI-RADS categorization or reliability of PSAD assessment as altering prostate volume [29,30,31].

Interestingly, logistic regression analysis identified the PI-RADS score as the sole independent imaging predictor of csPCa, while other MRI-derived variables such as prostate volume, lesion site, and lesion size did not demonstrate a distinct individual impact. However, this does not pose a problem for the DL model, where predictive interactions between features may be more complex and less easily disentangled. Unlike traditional statistical methods, DL architectures do not explicitly quantify the relative importance of each variable, instead capturing patterns within the data. Nonetheless, future studies could benefit from explainability techniques, allowing deeper insight into how imaging features, whether or not independently predictive in conventional analysis, influence the model’s risk assessment. This could extend to zonal locations of prostate lesions, such as the transition zone and peripheral zone, which were not evaluated in our cohort. Further areas for improvement include: (i) reducing the small but clinically significant number of high-grade cancers missed by M3, and identifying specific variables that could mitigate this limitation; and (ii) investigating whether the incorporation of biomarkers into initial risk stratification models could enhance patient selection for biopsy, as supported by recent findings in prostate cancer screening studies [32,33]; (iii) assessing how much limited inter-reader agreement of the PI-RADS [6] can influence the predictive performance of models using MRI data and which lines of score revision can improve the current scenario.

We must acknowledge several limitations of our study. First, the model was not externally validated on an independent dataset. Nonetheless, the performance of the M2 model (MRI-only) closely aligns with expected sensitivity and specificity benchmarks for prostate MRI, suggesting that the miscalibration we observed is an inherent limitation of the model rather than the effect of overfitting. However, M3 requires recalibration and validation in larger, diverse cohorts to confirm broader applicability. Second, we included Caucasian individuals only, suggesting that our models should be validated in a setting better reflecting different populations and related risk profiles for csPCa [34]. Third, the model did not assess the risk of ISUP 1 cancers. At this stage of research, this approach was intended to avoid overdiagnosis and the associated risk of overtreatment of indolent cancers, thereby enhancing the clinical relevance and safety of the model’s predictions.

Furthermore, DL models are characterized by some inherent limitation such as the limited ability to understand the inner workings of the adopted solution, furthermore compared to more traditional approaches they introduce a computational overhead that can affect their adoption in low-resources scenarios.

## 5. Conclusions

In our study, when applying a native DL approach for upfront risk prediction of csPCa, the model combining clinical and MRI-derived variables (M3) demonstrated the greatest utility in biopsy-averse scenarios by significantly reducing false positives while approaching benchmark sensitivity and NPV. However, the MRI-only model (M2) showed greater clinical utility in cancer-averse scenarios, offering higher sensitivity and net benefit at lower risk thresholds. These findings highlight the potential of DL-based models to support personalized risk stratification strategies and the potential stand-alone role for MRI, though external validation on larger and more diverse populations remains essential to confirm our results.

### Declaration of Generative AI and AI-Assisted Technologies in the Writing Process

During the preparation of this work, the authors used ChatGPT Free (OpenAI, San Francisco, CA, US) to revise English grammar and improve the overall text readability. After using this tool/service, the authors reviewed and edited the content as needed and take full responsibility for the intellectual and scientific content of the publication.

## Figures and Tables

**Figure 1 cancers-17-02257-f001:**
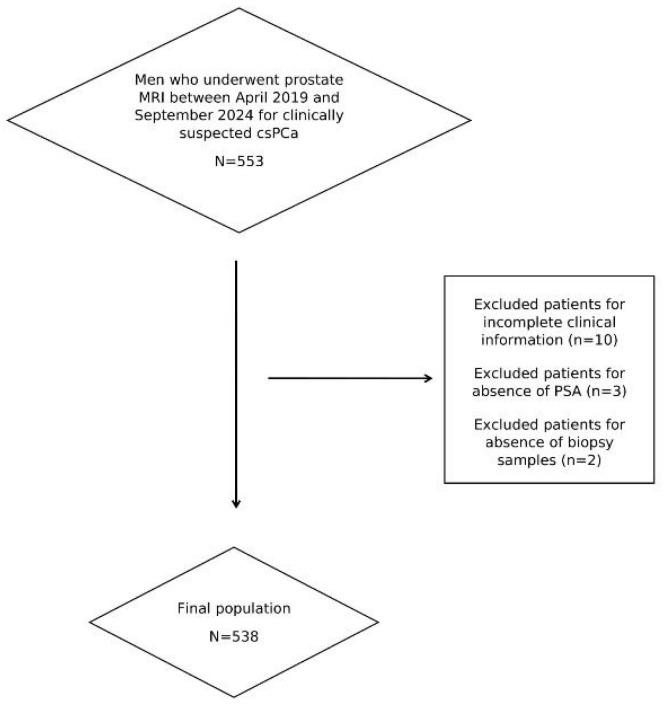
Study flowchart.

**Figure 2 cancers-17-02257-f002:**
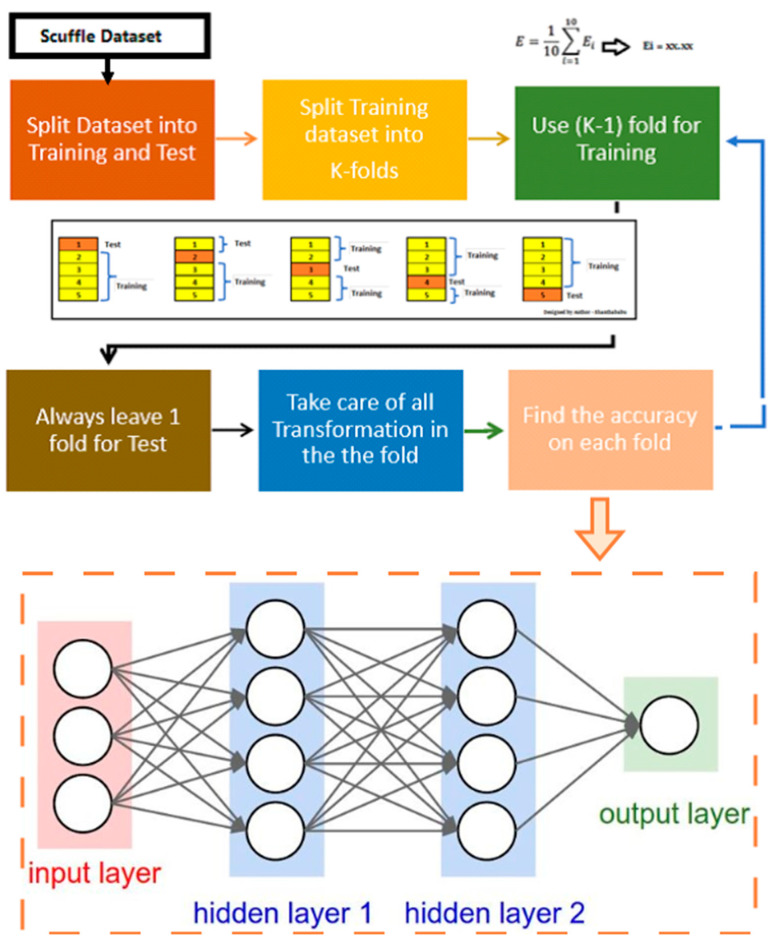
Pipeline for the construction of the deep learning model.

**Figure 3 cancers-17-02257-f003:**
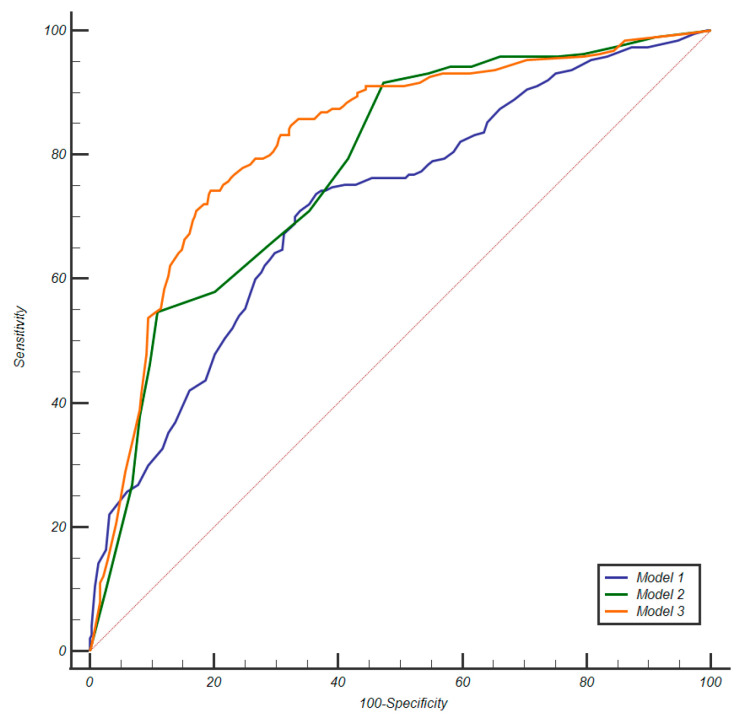
Comparison of AUCs for the three models. The pink diagonal line represents the reference line of no discrimination.

**Figure 4 cancers-17-02257-f004:**
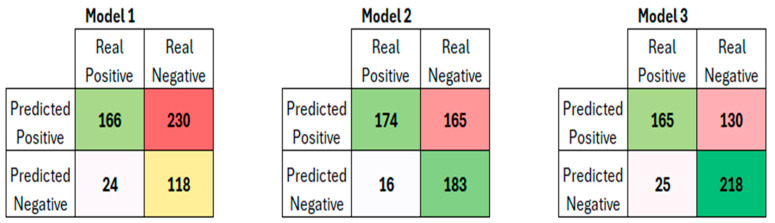
Confusion matrix for the three models. The cells coloured in green and yellow highlight the correct predictions (the more intense the green or the transition from yellow to green, the stronger the predictability). The cells coloured in white highlight the incorrect negative predictions, while the cells coloured in red highlight the incorrect positive predictions (the less intense the shading, the less severe the incorrect predictions).

**Figure 5 cancers-17-02257-f005:**
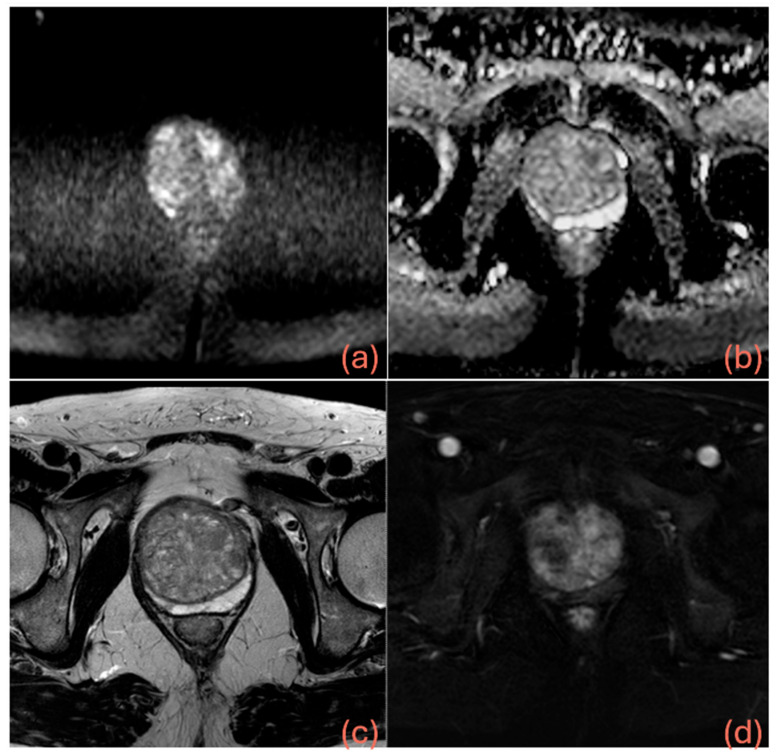
A 60-year-old man with PSA of 20 ng/mL, PSAD of 0.13 ng/mL/mL, negative digital rectal examination (DRE), no prior biopsy, and on tamsulosin therapy. MRI showed a prostate volume of 150 mL and revealed a 5 mm PIRADS 4 focal lesion in the right posteromedial peripheral zone at the apex—demonstrating hyperintensity on high b-value DWI (**a**), hypointensity on the ADC map (**b**) and T2-weighted images (**c**), and focal early enhancement on DCE (**d**). The lesion was classified as negative by model 1, positive by model 2, and negative by model 3. The latter classification could have potentially avoided an unnecessary biopsy, which was ultimately negative for prostate cancer on both targeted and perilesional sampling.

**Figure 6 cancers-17-02257-f006:**
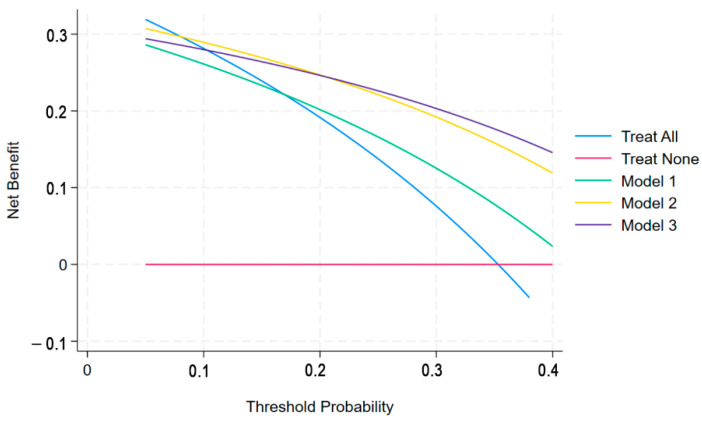
Decision-curve-analysis plots for the three models.

**Table 1 cancers-17-02257-t001:** Clinical, laboratory and imaging characteristics of the study population. Rates are calculated on the total of the included men (*n* = 432 for the training set and *n* = 108 for the test set), except for the site of index lesions (calculated on 378 men with a visible lesion for the training set and on 92 men for the test set) and the cancer stage (calculated 153 men with clinically significant prostate cancer for the training set e 37 men for the test set). DRE: digital rectal examination; GG: grade group; PI-RADS: Prostate Imaging Reporting and Data System; PSA: prostate-specific antigen; PSAD: prostate-specific antigen density.

	Training Set			Test Set		
	Median Value	Range, IQR	Rate	Median Value	Range, IQR	Rate
Age	67	43–84; (61–73)	-	66	49–80; (61–72)	-
PSA (ng/mL)	8.52	0.36–157; (4.3–9.3)	-	8.75	2–106; (4.9–8.9)	-
PSAD (ng/mL/mL)	0.15	0.002–1.98; (0.08–0.18)	-	0.17	0.03–2.6; (0.08–0.17)	-
Men with prior negative biopsy	-	-	93/430 = 0.22	-	-	25/108 = 0.23
Positive DRE	-	-	128/430 = 0.3	-	-	34/108 = 0.31
Family history of prostate cancer	-	-	76/430 = 0.18	-	-	16/108 = 0.15
Ongoing therapy for the prostate	-	-	164/430 = 0.38	-	-	45/108 = 0.41
Prostate volume (mL)	60	15–291; (38–73)	-	61.5	12–200; (37.75–80.5)	-
PI-RADS category of the index lesion						
PI-RADS 1			56/430 = 0.13			14/108 = 0.13
PI-RADS 2			41/430 = 0.10			12/108 = 0.11
PI-RADS 3			56/430 = 0.13			17/108 = 0.16
PI-RADS 4			159/430 = 0.37			41/108 = 0.38
PI-RADS 5			118/430 = 0.27			24/108 = 0.22
Site of visible index lesions on MRI						
Base			70/380 = 0.18			24/94 = 0.25
Mid-gland			161/380 = 0.43			40/94 = 0.43
Apex			80/380 = 0.21			15/94 = 0.16
			69/380 = 0.18			15/94 = 0.16
Size of MRI-visible lesions	13,7	4–47; (8–17)	-	12	4–35; (7–15)	-
Prostate cancer: histology						
No cancer	-	-	192/430 = 0.45	-	-	50/108 = 0.46
GG1	-	-	85/430 = 0.20	-	-	21/108 = 0.09
GG2	-	-	66/430 = 0.15	-	-	15/108 = 0.14
GG3	-	-	47/430 = 0.11	-	-	13/108 = 0.12
GG4	-	-	21/430 = 0.05	-	-	5/108 = 0.05
GG5	-	-	19/430 = 0.04	-	-	4/108 = 0.04
Prostate cancer: T stage on MRI						
≤T2			95/153 = 0.62			20/37 = 0.54
T3a			43/153 = 0.28			17/37 = 0.46
T3b			10/153 = 0.07			0
T4			5/153 = 0.03			0

**Table 2 cancers-17-02257-t002:** Results of logistic regression analysis. csPCa: clinically significant prostate cancer; DRE: digital rectal examination; MRI: magnetic resonance imaging; PI-RADS: Prostate Imaging Reporting and Data System; PSA: prostate-specific antigen; PSAD: prostate-specific antigen density.

Variable	Univariable Analysis			Multivariable Analysis	
	Prevalence in the Cohort (%)	Prevalence in Men with csPCa (%)	*p*	OR (95%CI)	*p*
Age ≥ 65 years	334/538 (62.1%)	141/190 (74.2%)	<0.0001	2.32 (1.42–3.50)	0.0005
PSA ≥ 10 ng/mL	111/538 (20.6%)	51/190 (26.8%)	0.0086	-	-
PSAD ≥ ng/mL/mL	196/538 (36.4%)	99/190 (52.1%)	<0.0001	2.70 (176–4.14)	<0.0001
Positive DRE	162/538 (30.1%)	78/190 (41%)	0.0002	1.73 (1.12–2.67)	0.0126
Prior negative biopsy	118/538 (21.9%)	25/190 (13.2%)	0.0003	0.43 (0.24–0.76)	0.0036
Ongoing therapy with alpha-blockers or 5-alpha reductase inhibitors	209/538 (38.8%)	64/190 (33.7%)	0.0697	-	-
Family history of csPCa	92/538 (17.1%)	27/190 (14.2%)	0.1888	-	-
Prostate volume on MRI			0.0018	-	-
1st quartile (12–38 mL)	138/538 (25.6%)	65/190 (34.2%)			
2nd quartile (39–52 mL)	137/538 (25.5%)	51/190 (26.8%)			
3rd quartile (53–73 mL)	129/538 (24%)	39/190 (20.5%)			
4th quartile (74–291 mL)	134/538 (24.9%)	35/190 (18.5%)			
Lesion site on MRI			0.0001	-	-
No visible lesions	65/538 (11.4%)	5/190 (2.6%)			
Base	105/538 (19.6%)	39/190 (20.6%)			
Mid-gland	214/538 (39.8%)	79/190 (41.6%)			
Apex	97/538 (18.4%)	38/190 (20%)			
More sites involved	57/538 (10.8%)	29/190 (15.2%)			
Lesion size on MRI			<0.0001	-	-
Non visible	65/538 (12.1%)	5/190 (2.6%)			
<1 cm	173/538 (32.2%)	38/190 (20%)			
1–1.9 cm	217/538 (40.3%)	96/190 (50.5%)			
2–2.9 cm	54/538 (10%)	27/190 (14.2%)			
≥3 cm	29/538 (5.4%)	24/190 (12.7%)			
PI-RADS category			<0.0001	10.32 (5.83–18.27)	<0.0001
1–3	199/538 (37%)	16/190 (8.4%)			
4–5	339/538 (63%)	174/190 (91.6%)			

**Table 3 cancers-17-02257-t003:** Performance of the three models in discriminating clinically significant prostate cancer (grading group ≥ 2). AUC = area under the curve; FN = false negatives; FP = false positives; NPV = negative predictive value; PI-RADS = Prostate Imaging Reporting and Data System; PPV = positive predictive value; TN = true negatives; TP = true positives; 95% CI = 95% confidence intervals.

Model	AUC (95%CI)	Threshold	Sensitivity % (95%CI)	Specificity % (95%CI)	NPV % (95%CI)	PPV % (95%CI)	ISUP Grading Group (GG) of Missed csPCa
Model 1	0.716 (0.676–0.754)	Confidence ≥ 0.27	87.4 (81.8–91.7)	33.9 (28.9–39.1)	83.1 (84.7–92.7)	41.9 (39.6–44.2)	14/24 GG2 6/24 GG3 1/24 GG4 3/24 GG5
Model 2	0.778 (0.740–0.812)	Confidence ≥ 0.21, corresponding to PI-RADS ≥ 4	91.6 (86.6–95.1)	52.6 (47.2–57.9)	91.9 (87.6–94.8)	51.3 (48.3–54.3)	13/24 GG2 3/24 GG3
Model 3	0.822 (0.787–0.853)	Confidence ≥ 0.32	86.8 (81.2–91.3)	62.6 (57.3–67.7)	89.7 (85.7–92.7)	55.9 (52.2–59.5)	19/24 GG2 6/24 GG3

**Table 4 cancers-17-02257-t004:** Net benefit of the biopsy strategies in detecting clinically significant prostate cancer, which was assumed to be grading group ISUP (International Society of Urological Pathology) 2 or larger. csPCa = clinically significant prostate cancer.

Biopsy Strategies		Net Benefit			
	10%	15%	20%	25%	30%
Model 1	0.261	0.233	0.202	0.166	0.125
Model 2	0.289	0.269	0.247	0.221	0.192
Model 3	0.280	0.264	0.246	0.226	0.203
Treat all (biopsying any lesion)	0.281	0.239	0.191	0.138	0.076
Treat none (biopsying no lesions)	0	0	0	0	0

## Data Availability

The data that support the findings of this study are available from the corresponding author upon reasonable request.

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
