# Peer review of "A Deep Learning Model Integrating Clinical and MRI Features Improves Risk Stratification and Reduces Unnecessary Biopsies in Men with Suspected Prostate Cancer"

_cancers, 2025, doi:10.3390/cancers17132257_

Round 1

Reviewer 1 Report

Comments and Suggestions for Authors

The authors used deep-learning techniques to improve the risk stratification of suspected prostate cancer using clinical and MRI-based features. They tested 3 models, including one solely relying on clinical features, one mainly focusing on MRI-based features, and the last one combining both clinical and MRI-based features. They revealed that the last model performed the best in terms of Area under the Curve (AUC) value, and found that the model can reduce unnecessary biopsies by 43.4% compared to the first model and 21.2% compared to the second model. However, there are some concerns regarding the models evaluated in the study. It is uncommon to screen many different machine learning models to obtain the best model, but the authors only conducted the analysis with 3 models. The following list out other potential issues:

  1. Abbreviations in the simple summary and abstract should be provided with full name when first described.
  2. It is not clear what kind of clinical variables were used by the model 1 in the abstract.
  3. The AUC was only reported for the model 3. For better understanding and comparison, AUCs for the other two models should be described in the abstract.
  4. The performances for these 3 models seems to be not compared adequately, as details of F1 value are required to be provided.
  5. The conclusion for combining clinical and MRI features in model 3 to improve csPCa risk stratification is not convincing, as M3 has slightly lower sensitivity than M2. The authors also didn’t give enough description about the sensitivity using an exact numeric value. Thus it is hard to believe the conclusion.
  6. A second cohort to validate the performance of these 3 models is strongly recommended.
  7. In Figure 2, the workflow has a typo at the bottom of the left part, which is “Take care of all Transformation in the the fold.”
  8. Whenever possible, please provide adequate references for these algorithms used by the deep learning method used by current pipeline. For example, the introduction of Binary Cross-Entropy (BCE) loss function can be further elaborated, and the description of Adam optimizer should be provided with appropriate citations.
  9. In terms of PPV and NPV, details about their calculations are required. Again, AUC comparisons across 3 models using the DeLong method should be added with appropriate citations.
  10. The sentence describing miss rate for csPCa at page 8 is not completed, as it intends to explain that 13.2% miss rate is observed for model 3 (M3) but the authors forgot the add the target ‘M3’ at the end of the sentence.
  11. In Table 2, it is unclear why there is only one p value for the comparisons among different quartile of prostate volume on MRI against the 1st The same question is applicable to other MRI features included in the Table 2.
  12. Table 3 displays the moderate differences in terms of AUCs across 3 models, and these 3 models do not perform impressively given its relative low AUCs values. It is unclear why the authors did not try other machine learning models to evaluate the performances, such as random forest, SVM, decision tree, and many others. The screening for the best model seems to be too limited among 3 models in the current manuscript.
  13. Germline or somatic mutations, such as mutations on TMPRSS2 gene, in blood sample, can be used as diagnostic markers for csPCa, but the authors neither mentioned them in the introduction nor collected potential mutations for training the models for csPCa.

Author Response

Reviewer #1:

  1. The authors used deep-learning techniques to improve the risk stratification of suspected prostate cancer using clinical and MRI-based features. They tested 3 models, including one solely relying on clinical features, one mainly focusing on MRI-based features, and the last one combining both clinical and MRI-based features. They revealed that the last model performed the best in terms of Area under the Curve (AUC) value, and found that the model can reduce unnecessary biopsies by 43.4% compared to the first model and 21.2% compared to the second model. However, there are some concerns regarding the models evaluated in the study. It is uncommon to screen many different machine learning models to obtain the best model, but the authors only conducted the analysis with 3 models.

Please, see the reply at point # 13b below.

  1. Abbreviations in the simple summary and abstract should be provided with full name when first described.

Thank you for noticing. We opened all the abbreviations and revised the abstract in accordance with points #3, #4, and #6 while remaining within the journal’s word limit.

  1. It is not clear what kind of clinical variables were used by the model 1 in the abstract.

Please, see point # 2 above.

  1. The AUC was only reported for the model 3. For better understanding and comparison, AUCs for the other two models should be described in the abstract.

Please, see point # 2 above.

  1. The performances for these 3 models seems to be not compared adequately, as details of F1 value are required to be provided.

We are not certain we fully understood this point. The F1 score is particularly useful in scenarios with class imbalance, such as low disease prevalence (Park HS et al., Radiol Med 2024;129:1644–1655). However, our cohort exhibited a clinically significant prostate cancer (csPCa) prevalence of 35.3%, which falls within the expected range of 30–50%. Moreover, the F1 score is derived from true positives, false positives, and false negatives, and may therefore be redundant in the context of the already reported performance metrics—sensitivity, specificity, positive predictive value (PPV), negative predictive value (NPV), and net benefit—which are widely accepted and representative of expected outcomes in prostate MRI literature (e.g., see reference #11 regarding the Mount Sinai deep learning-based upfront risk calculator).

So, while we calculated the F1 scores for each model (Model 1: 0.457, Model 2: 0.495, Model 3: 0.504), we ultimately chose not to include them in the paper, as they could be misleading. These low values primarily reflect the inherent trade-off between sensitivity and specificity — a well-known challenge in prostate MRI interpretation. The aim of this study was not to develop an unrealistic “perfect” classification tool that balances high sensitivity and high specificity. Rather, our goal was to propose a practical, real-world solution to reduce false positives by working on the specificity/PPV side. This work does not claim to offer a definitive endpoint, but rather serves as a contribution to identify the right path toward progressive improvements in our upfront risk stratification models.

We believe that, in this context, the F1 score alone does not fully capture the clinical value of the models. Instead, the reported performance metrics more clearly demonstrate that Model 3 contributes meaningfully to addressing an unmet clinical need—namely, reducing unnecessary prostate biopsies by minimizing false-positive findings.

  1. The conclusion for combining clinical and MRI features in model 3 to improve csPCa risk stratification is not convincing, as M3 has slightly lower sensitivity than M2. The authors also didn’t give enough description about the sensitivity using an exact numeric value. Thus it is hard to believe the conclusion.

Please, see point # 2 above.

  1. A second cohort to validate the performance of these 3 models is strongly recommended.

We agree on the importance of external validation; however, including it at this stage would represent a post-hoc deviation from the original study design. While we acknowledge its necessity and are actively pursuing it, external validation remains challenging due to the lack of publicly available or institutionally accessible datasets with the required clinical and MRI detail for the very specific task we aimed at. Nonetheless, we have addressed this point in the Discussion, noting how the alignment of our findings with expected prostate MRI performance supports their potential reproducibility.

  1. In Figure 2, the workflow has a typo at the bottom of the left part, which is “Take care of all Transformation in the the fold.”

Thank you for pointing that out. We made the correction as suggested.

  1. Whenever possible, please provide adequate references for these algorithms used by the deep learning method used by current pipeline. For example, the introduction of Binary Cross-Entropy (BCE) loss function can be further elaborated, and the description of Adam optimizer should be provided with appropriate citations.

Thank you for the suggestion. We added a paragraph for both the BCE and Adam Optimizer, providing a more detailed description of their functioning. We also added appropriate references for each technique (new references # 17-18).

  1. In terms of PPV and NPV, details about their calculations are required. Again, AUC comparisons across 3 models using the DeLong method should be added with appropriate citations.

We clarified the calculation of PPV and NPV by adding the following sentence to the Materials and Methods, paragraph 2.6 on “Data analysis”: “Positive Predictive Value (PPV) and Negative Predictive Value (NPV) were calculated by classifying each case as ‘positive’ or ‘negative’ according to whether the model output was above or below the confidence threshold identified through ROC analysis, respectively. The calculations followed standard formulas as reported elsewhere [19]”. We chose not to report the well-known formulas directly, but instead cited a contemporary source that discusses the application of diagnostic performance metrics in the context of AI, which we believe may serve as a valuable resource for readers (Park HS et al., Radiol Med 2024;129:1644–1655).

We included a reference (new # 20) to support the use of the DeLong method for ROC curve comparison (Obuchowski NA, Fundamentals of Clinical Research for Radiologists: ROC Analysis. AJR Am J Roentgenol. 2005;184(2):364–372) and updated the numbering of all the references accordingly.

  1. The sentence describing miss rate for csPCa at page 8 is not completed, as it intends to explain that 13.2% miss rate is observed for model 3 (M3) but the authors forgot the add the target ‘M3’ at the end of the sentence.

Thank you for pointing that out. We have now included the missing “for M3” in the previously incomplete sentence in paragraph 3.2.

  1. In Table 2, it is unclear why there is only one p value for the comparisons among different quartile of prostate volume on MRI against the 1stThe same question is applicable to other MRI features included in the Table 2.

Thank you for your thoughtful question. We appreciate the opportunity to clarify this point.

In Table 2, the p-values reported are not the result of pairwise comparisons between quartiles of prostate volume or other MRI features. Rather, they reflect the results of univariable logistic regression analyses, where each variable—whether ordinal (e.g., quartiles of prostate volume) or dichotomous—is assessed in relation to the binary outcome variable (biopsy positive vs. biopsy negative).

Specifically, the p-values were derived using the chi-square test, as implemented in our statistical software package (MedCalc). This approach evaluates whether the distribution of cases across the categories of each variable differs significantly between the two outcome groups, as expected in univariable/multivariable analysis. It does not perform multiple pairwise comparisons between individual quartiles or categories.

In the case of prostate volume, the analysis showed a significantly higher proportion of cases in the biopsy-negative group across all quartiles, which is reflected in the single p-value reported. This same principle applies to all other variables in the table, including dichotomous ones.

For further clarity, we have included a screenshot from the software output illustrating how the test was conducted (see attached image).

  1. Table 3 displays the moderate differences in terms of AUCs across 3 models, and these 3 models do not perform impressively given its relative low AUCs values. It is unclear why the authors did not try other machine learning models to evaluate the performances, such as random forest, SVM, decision tree, and many others. The screening for the best model seems to be too limited among 3 models in the current manuscript.

a). Regarding the AUC values, we acknowledge that they are relatively low; however, they should be interpreted within the well-known context of the limited specificity of prostate MRI and clinically based upfront risk stratification as presented in the Introduction. In such a setting, it would be unscientific to aim for artificially “impressive” AUCs, given the unavoidable trade-off between sensitivity and specificity. Since the current objective is to improve specificity, we believe our study reasonably demonstrated that an AI-based clinico-radiological risk calculator can offer tangible benefits, while also outlining future directions for research. To address the reviewer’s observation, we have added the following sentence to the first paragraph of the Discussion section: “While the relatively low AUCs of the models, including M3, still reflect room for further improvement in specificity and the unavoidable trade-off with sensitivity, we believe our work provides a research direction in which AI-based calculators can meaningfully enhance biopsy decision-making compared to the current clinical scenario.”

b). Regarding the second comment, about the selection of multiple models for the comparison, we agree that it is typical in machine learning oriented papers, however the goal of this work is to use machine learning models as a mean to explore different combinations of features to better understand which ones are more reliable predictors of csPCa. Therefore we kept the architecture consistent to avoid the introduction of external factors in the analysis.

  1. Germline or somatic mutations, such as mutations on TMPRSS2 gene, in blood sample, can be used as diagnostic markers for csPCa, but the authors neither mentioned them in the introduction nor collected potential mutations for training the models for csPCa.

Thank you for this very relevant observation. We fully agree that germline and somatic mutations—such as those involving the TMPRSS2 gene—are of growing interest as potential diagnostic markers for csPCa. However, our study focuses on the early diagnostic setting (men with clinical suspicion of csPCa with cancer prevalence 30-50%), where genetic testing is not yet routinely implemented given cost-effectiveness considerations. This is why we believe mentioning this in the Introduction could be redundant.

Genetic markers have shown promise in the different setting of prostate cancer screening—most notably in the STHLM3 trial (Eklund M et al., Ne Engl J Med 2021;385:908-920), which incorporated a genomic risk score. However, they remain a costly solution compared to other, more accessible biomarkers. For example, the 4Kscore panel, used in the PROSCREEN trial (Auvinen A et al., JAMA 2024;331:1452-1459), has demonstrated effective upfront risk stratification prior to MRI and prostate biopsy, contributing to avoid many unnecessary biopsies.

To account for the reviewer’s suggestion, we have refined the following statements in the Discussion section: “Further areas for improvement include: (i) reducing the small but clinically significant number of high-grade cancers missed by M3, and identifying specific variables that could mitigate this limitation; and (ii) investigating whether the incorporation of biomarkers into initial risk stratification models could enhance patient selection for biopsy, as supported by recent findings in prostate cancer screening studies [30-31]”.

Reviewer 2 Report

Comments and Suggestions for Authors

I have reviewed your study titled "A Deep Learning Model Integrating Clinical and MRI Features Improves Risk Stratification and Reduces Unnecessary Biopsies in Men with Suspected Prostate Cancer" in detail. I have listed the points I found missing in the article. Removing the specified deficiencies will increase the quality of the article. Too many numerical values ​​are presented in the abstract section. In this section, the reason for the study, its contributions to the literature, and its innovative aspects should be highlighted. Information about the models should be provided. Numerous studies are available in the literature on this subject. It is important to expand these studies. "Deep learning-based PI-RADS score estimation to detect prostate cancer using multiparametric magnetic resonance imaging". A paragraph about the organization of the article should be added at the end of the introduction section. Sample images from the dataset, the number of data in each class, etc., should be presented. More information about the dataset is required. Models 1, 2, and 3 should be detailed by supporting them with figures. If possible, the application results should be supported with a confusion matrix, etc. Different metrics can be used to evaluate the performance of these models. E.g., Accuracy, sensitivity, F1-score, etc. Please clearly explain the innovative aspects of the article in the last paragraph of the Introduction section. It is pretty challenging to understand the innovative aspect of the article.

Author Response

Reviewer #2:

I have reviewed your study titled "A Deep Learning Model Integrating Clinical and MRI Features Improves Risk Stratification and Reduces Unnecessary Biopsies in Men with Suspected Prostate Cancer" in detail. I have listed the points I found missing in the article. Removing the specified deficiencies will increase the quality of the article.

  1. Too many numerical values ​​are presented in the abstract section. In this section, the reason for the study, its contributions to the literature, and its innovative aspects should be highlighted. Information about the models should be provided.

  1. Numerous studies are available in the literature on this subject. It is important to expand these studies. "Deep learning-based PI-RADS score estimation to detect prostate cancer using multiparametric magnetic resonance imaging".

Thank you for suggesting the article titled "Deep learning-based PI-RADS score estimation to detect prostate cancer using multiparametric magnetic resonance imaging." Unfortunately, we were unable to find this publication in either PubMed or Scopus. Should you be able to provide the full reference, we would be pleased to reconsider its relevance to our work.

We respectfully disagree with the broader assertion that this area is already well represented in the literature. While numerous DL-based studies exist, the very large majority focus on classification or detection tasks directly applied to prostate MRI images. Similarly, most risk calculators are based on conventional statistical models, such as logistic regression, and do not incorporate DL methodologies.

Despite a second, careful review of the literature, we were unable to identify additional studies that align with our specific focus: DL-based risk calculators that integrate both clinical and MRI-derived variables. The only recent study we identified that used MRI data alone to predict outcomes (Roest C et al., Radiology: Imaging Cancer 2025;7(1):e240078) addresses a distinctly different objective, namely the prediction of progression from low- to high-risk prostate cancer. As such, it was deemed not directly relevant to the scope of our study and was therefore not included.

Notably, our models did not use MRI images directly, but rather structured data derived from MRI reports—such as the PI-RADS category (Model 2), and additional information including prostate volume, lesion site, and lesion size (Model 3). This design was intentional, as we know better stated at the end of the Introduction as follows: “The innovation of our study lies in applying deep learning for upfront risk stratification—rather than relying on traditional logistic regression models commonly used in non–DL-based calculators—and in leveraging structured MRI-derived data without requiring direct image analysis or the use of DL systems for interpreting MRI images. This approach enhances accessibility and usability, especially for clinicians who do not read MRI images or do not have access to AI tools for image interpretation”. Please, se also reply to point # 9 by reviewer # 2 for this addition.

  1. A paragraph about the organization of the article should be added at the end of the introduction section.

Thank you for the suggestion. We placed the following statement at the end of the Introduction: “This article is organized into sections detailing the methodology, dataset construction, and model development, followed by validation results and discussion of clinical implications”.

  1. Sample images from the dataset, the number of data in each class, etc., should be presented.

We fully agree on the importance of providing a detailed description of the dataset. However, as the complete dataset characteristics are already presented in Table 2, we are uncertain whether we have fully understood the concern raised. Regarding imaging data, we have now included a figure illustrating a representative case with corresponding MRI images to enhance clarity.

  1. More information about the dataset is required.

Unfortunately, we were unable to understand the level of detail expected by the reviewer, or which additional variables might be considered necessary beyond those already presented in Tables 1 and 2, where the main clinical and MRI characteristics of the cohort are comprehensively detailed.

On this basis, we could not take specific action, but we remain available to provide further clarification or additions should the reviewer offer more precise guidance.

  1. Models 1, 2, and 3 should be detailed by supporting them with figures.

The difference between the three models adopted in the paper lies in the feature selected for the analysis and not in their architecture; therefore, we are afraid that including additional figures to describe them wouldn’t provide any additional information. We remain fully open to incorporating additional visual material should the Editor deem it necessary.

  1. If possible, the application results should be supported with a confusion matrix, etc.

While the predicted versus observed values were already available in Table 3 as true positives, false positives, true negatives, and false negatives, we acknowledge that presenting the same data in a confusion matrix format may offer a more intuitive and visually appealing summary. We have therefore included this as new Figure 4. Tab. 3 was revised accordingly to avoid redundancies.

  1. Different metrics can be used to evaluate the performance of these models. E.g., Accuracy, sensitivity, F1-score, etc.

Thank you for the thoughtful question. We acknowledge that additional performance metrics can be used to evaluate predictive models. The F1-score, in particular, is often recommended in settings with significant class imbalance, as noted by Park HS et al. (Radiol Med 2024;129:1644–1655). However, in our cohort, the prevalence of clinically significant prostate cancer (csPCa) was 35.3%, which falls within the expected range of 30–50% and does not represent a highly imbalanced scenario. Regarding sensitivity, we are unsure whether we fully understood the reviewer’s concern, as this metric is openly reported in Tab. 3.

More importantly, the core clinical challenge addressed in our study is the reduction of false positives, which can lead to unnecessary biopsies. For this reason, we prioritized performance metrics that directly reflect this clinical problem—namely, specificity, PPV, and net benefit as deriving from decision curve analysis. These metrics are well-established in prostate MRI literature and provide a concise assessment of model performance in this context, avoiding redundancies from additional metrics that, while significant per se, do not add substantial information. E.g., it is doubtful what “accuracy” values can add to AUCs or the other metrics once the balance between true-positives and false-positives has been put into the spotlight.

Assuming that the value of a study lies in how effectively its results emphasize clinical relevance rather than the sheer number of performance metrics reported, we took the liberty of not making changes in this regard. However, we remain fully open to further discussion on this.

Please, see also reply to point # 5 by reviewer # 1 on this topic.

  1. Please clearly explain the innovative aspects of the article in the last paragraph of the Introduction section. It is pretty challenging to understand the innovative aspect of the article.

We acknowledge the need for emphasizing this, and added the following statements at the end of the Introduction: “The innovation of our study lies in applying deep learning for upfront risk stratification—rather than relying on traditional logistic regression models commonly used in non–DL-based calculators—and in leveraging structured MRI-derived data without requiring direct image analysis or the use of DL systems for interpreting MRI images. This approach enhances accessibility and usability, especially for clinicians who do not read MRI images or do not have access to AI tools for image interpretation”.

Reviewer 3 Report

Comments and Suggestions for Authors

The article addresses an important clinical question and demonstrates innovation. However, it requieres clarifications, I describe as follow:

Abstract: State the time period of data collection. Caucasian should be moved from the abstract

Introduction: Introduce specific hypothesis. Explain more limitations of DL models

Methods: Define "biopsy-averse and cancer averse" scenarios. Explain more the lacking of external validation

Results: Did the authors analysed PSA levels with other variables? Even the model performance improve (p value), is it clinically meaningful in all settings?

Discussion: Discuss more the potential overfitting, despite dropout and weight decay

Recommend potential for future explainable AI integration

Discuss PIRADS inter-reader variability even with AI

Author Response

Reviewer #3:

1.State the time period of data collection. Caucasian should be moved from the abstract

Thank you for your suggestion. We have thoroughly revised the Abstract in response to the other reviewers’ observations, incorporating your recommended changes as well.

2.Introduce specific hypothesis.

Thank you for this observation. We are not entirely certain we fully understood the concern, as we believe a specific hypothesis was indeed introduced, namely, whether DL-based models integrating clinical and MRI-derived variables can contribute to reducing unnecessary prostate biopsies in men with suspected prostate cancer. We hope that the recent expansion of the final paragraph of the Introduction, made in response to previous reviewer comments, further clarifies this point by highlighting both the hypothesis and the innovative aspects of our study design.

  1. Explain more limitations of DL models

Thank you for the valuable suggestion. We introduced an additional paragraph in the discussion section describing the some typical limitations of DL models, especially in the context of medical applications, such as the limited explainability and their computational overhead compared to traditional ML models.

4.Define "biopsy-averse and cancer averse" scenarios.

To enhance clarity around these decision-making frameworks, we have revised the second paragraph of the Discussion section as follows: “DCA underscored the complementary utility of our models across varying clinical priorities. M2 achieved greater net benefit at threshold probabilities ≤20%, reflecting a cancer-averse scenario wherein the primary objective is minimizing missed csPCa diagnoses, even at the expense of increased biopsy rates. Conversely, M3 outperformed above this threshold, aligning with a biopsy-averse perspective that favors reducing unnecessary procedures while maintaining acceptable diagnostic sensitivity. These findings suggest that M2 and M3 may be selectively applied to support personalized decision-making, depending on the clinical scenario and the shared preference between patient and clinician [23]”.

  1. Explain more the lacking of external validation

We agree on the importance of external validation; however, including it at this stage would represent a post-hoc deviation from the original study design. While we acknowledge its necessity and are actively pursuing it, external validation remains challenging due to the lack of publicly available or institutionally accessible datasets with the required clinical and MRI detail for the very specific task we aimed at.

Nonetheless, we have addressed this point in the Discussion, noting how the alignment of our findings with expected prostate MRI performance supports their potential reproducibility.

6.Did the authors analysed PSA levels with other variables?

As detailed in Paragraph 2.3, Model 1 was based solely on clinical variables (variables 1–7), including age, PSA, PSA density, DRE findings, family history, prior negative biopsy, and ongoing therapy. Model 2 incorporated imaging features from prostate MRI, specifically the PI-RADS 2.1 classification system. Model 3 combined both clinical and MRI-derived variables (variables 1–11), including lesion size, location, and prostate volume.

As implicitly noted by the reviewer, the inclusion of more variables generally enhances predictive performance. We believe we have selected variables that are readily accessible in routine clinical practice.

  1. Even the model performance improve (p value), is it clinically meaningful in all settings?

We reported the p-value for AUC comparison between models in the Results section. In particular, we demonstrating that M3 achieved a significantly higher AUC than M2 (p = 0.0003). However, we concur with Vickers et al. (Vickers AJ, van Calster B, Steyerberg EW. A simple, step-by-step guide to interpreting decision curve analysis. Diagn Progn Res. 2019;3:18) that statistical significance alone does not equate to clinical utility. For this reason, we performed decision curve analysis to assess and compare the net benefit of the models across a range of threshold probabilities. As detailed in the Discussion section, this approach provided a more clinically relevant evaluation of model performance.

8.Discuss more the potential overfitting, despite dropout and weight decay

 Thank you for your suggestion, while we agree that overfitting can be a critical concern in low-data scenarios, we believe that the architectural choices adopted in the present work, namely: a relatively small NN architecture, an aggressive dropout strategy and the iontrodcution of a weight decay parameter, effectively mitigated this issue. This is confirmed by our experimental results where, even by adopting a k-fold cross validation setup, the model still exhibits strong generalization capabilities to the test data at each iteration.

9.Recommend potential for future explainable AI integration

Thank you for raising this important point. We believe it has already been addressed in the Discussion, as follows: “Nonetheless, future studies could benefit from explainability techniques, allowing deeper insight into how imaging features, whether or not independently predictive in conventional analysis, influence the model’s risk assessment”.

10.Discuss PIRADS inter-reader variability even with AI

This is a very interesting and relevant point. To the best of our knowledge, no previous studies have specifically investigated the interplay between PI-RADS inter-reader variability and AI-based models—whether in terms of model training or their influence on clinical decision-making.

Nonetheless, we acknowledge that limited inter-reader agreement in PI-RADS scoring could impact the performance and generalizability of models relying on MRI-derived inputs. To address this, we have expanded the Discussion to include the following as a future research priority: “Further areas for improvement include [...] assessing how much limited inter-reader agreement of the PI-RADS [32] can influence the predictive performance of models using MRI data and which lines of score revision can improve the current scenario”. We added new reference 32 to support this.

Round 2

Reviewer 2 Report

Comments and Suggestions for Authors

Congratulations on the successful revision.